# Genetic Improvement of *Arundo donax* L.: Opportunities and Challenges

**DOI:** 10.3390/plants9111584

**Published:** 2020-11-16

**Authors:** Tommaso Danelli, Marina Laura, Marco Savona, Michela Landoni, Fabrizio Adani, Roberto Pilu

**Affiliations:** 1Gruppo Ricicla Labs—Department of Agricultural and Environmental Sciences—Production, Landscape and Agroenergy, Università’ Degli Studi di Milano, Via Celoria 2, 20133 Milan, Italy; tommaso.danelli@unimi.it (T.D.); fabrizio.adani@unimi.it (F.A.); 2Agricultural Genetics Group—Department of Agricultural and Environmental Sciences—Production, Landscape and Agroenergy, Università’ Degli Studi di Milano, Via Celoria 2, 20133 Milan, Italy; 3CREA, Research Centre for Vegetable and Ornamental Crops, Corso Degli Inglesi 508, 18038 Sanremo, Italy; marina.laura@crea.gov.it (M.L.); marco.savona@crea.gov.it (M.S.); 4Department of Biosciences, Università’ Degli Studi di Milano, Via Celoria 26, 20133 Milan, Italy; michela.landoni@unimi.it

**Keywords:** *Arundo donax*, clonal selection, somaclonal variation, in vitro culture, genetic improvement

## Abstract

*Arundo donax* L., the giant reed—being a long-duration, low-cost, non-food energy crop able to grow in marginal lands—has emerged as a potential alternative to produce biomass for both energy production, with low carbon emissions, and industrial bioproducts. In recent years, pioneering efforts have been made to genetically improve this very promising energy crop. This review analyses the recent advances and challenges encountered in using clonal selection, mutagenesis/somaclonal variation and transgenesis/genome editing. Attempts to improve crop yield, in vitro propagation efficiency, salt and heavy metal tolerance by clonal selection were carried out, although limited by the species’ low genetic diversity and availability of mutants. Mutagenesis and somaclonal variation have also been attempted on this species; however, since *Arundo donax* is polyploid, it is very difficult to induce and select promising mutations. In more recent years, genomics and transcriptomics data are becoming available in Arundo, closing the gap to make possible the genetic manipulation of this energy crop in the near future. The challenge will regard the functional characterization of the genes/sequences generated by genomic sequencing and transcriptomic analysis in a complex polyploid genome.

## 1. Introduction

*A. donax* L. is a widespread species of unclear origin. This perennial grass grows spontaneously in temperate and tropical zones almost all over the world [1,2]. It can be found in ecosystems highly altered by anthropic activity and along riparian zones [3], where it often acts as an invasive weed reducing biodiversity [4], and brings an increased risk of wildfires and floods [5]. The roots can grow to 5 m in depth [1,6] and canes can reach 8–10 m in height and 3–4 cm in diameter (Figure 1). The leaves are flat, 5–8 cm wide and 30–100 cm long, inserted alternately in two ranks [1,7,8]. In southern Europe, new canes sprout continuously from rhizomes, starting in early March until August to November, when flowering takes place. Senescence follows in winter, with canes becoming yellow and generally losing leaves and inflorescences. Inflorescences are large plume-like panicles 30–100 cm long [7] that do not produce viable seeds [9,10,11,12,13].

Studies on *A. donax* L. sterility are often contradictory. In fact, this topic has yet to be clarified, since a drastic founder effect could explain this, rather than it being a consequence of defective chromosome pairing in aneuploid *A. donax* and *A. micrantha* Lam. [14]. At first glance, in *A. donax*, male and female gametogenesis fails right after meiosis. Following the megaspores’ mother cell formation at the tetrad stage, three chalazal megaspores degenerate, while one micropilar megaspore enlarges, develops a large nucleolus but no embryo sac, with the consequent proliferation of dysfunctional cells and the failure of ovule development. Pollen grains’ cell walls usually collapse by autolysis, with the appearance of large numbers of vacuoles and variable numbers of nuclei and micronuclei. Despite this being the common result, the formation of a few viable pollen grains is reported with a frequency of 6.2% [14,15]. Meiosis occurs in less than 10% of microsporocytes, and no formation of exine occurs in the microsporangium [11]. *A. donax* sterility has been reported to be related to alterations in gametogenesis and fertilization and post-fertilization development [16]. *A. donax* sterility most likely has various causes that have led to its agamic propagation strategy. Reproduction is exclusively asexual and occurs through vegetative propagation [17,18] by fragmentation of rhizomes and cane fragments, which are dispersed by floods or by human activity [1,19].

The worldwide spread of *A. donax* is related to several domestic and agricultural purposes such as the making of walking-sticks, baskets, mats, fishing rods, fences, plant stakes and musical instruments’ parts, especially the reeds for clarinets and saxophones [1,8,20,21]. *A. donax* is supposed to have spread from Asia, its native center, to America, passing through the Mediterranean area [15,22,23,24,25]. Other authors suggest that this plant originated in Mediterranean regions from native species [26]. At least four other species from the *Arundo* taxon are present in the Mediterranean area: *A. plinii* L., *A. collina* Ten., *A. mediterranea,* and *A. micrantha* Lam. [27,28]. Six lineages of *A. donax* are supposed to be distributed from Asia to the Mediterranean basin, with a putative area of origin in the Western and Southern edges of the Qinghai-Tibet Plateau [29]. The phylogenesis of *A. donax* is still debated, as the hypothesis that this species is polyploid or allopolyploid is shared by various authors based on its macroscopic traits, such as the great vegetative vigor and the absence of fertile seeds [12,13,30]. The literature data about the chromosome number of *A. donax* L. show some discrepancies, from an often-reported number of 108–110 chromosomes [12,30,31,32,33], to 84 chromosomes [13], or seed-producing cytotypes with 72 chromosomes, although this last result was published before the revised classification of the genus *Arundo* [28].

Large-scale cultivation of *A. donax* was established between the 1930s and the 1960s in Northern Italy to produce the textile fiber Rayon, but after the Second World War, it became unsustainable due to the competition from petroleum-derived products [1,34]. Recently, the economic interest in this species has risen again and a considerable number of publications have dealt with different topics about this interesting crop: bioenergy, agronomy, invasiveness, and its use for phytoremediation [35]. The high yield in dry matter per hectare and the low inputs required for cultivation make *A. donax* a promising energy crop [8,36,37]. Chips and pellets for direct combustion are a practical solid fuel obtainable from *A. donax* [38], while other possibilities are gasification [39], second-generation bioethanol [40,41], and biogas for co-generation. For the last-named use, various experiments in northern Italy have promoted *A. donax* as an acceptable substitute of *Zea mays* L. in anaerobic digestion plants. Other potential benefits include biofuels [42,43], biocompounds for plastic polymers [44], green building technologies [45] and leaf protein concentrate as a feed ingredient [46].

The capacity of *A. donax* to grow on marginal and abandoned lands makes this plant viable for cultivation on soil not suitable for traditional agriculture. Lands can be defined as marginal for different reasons, such as water scarcity, poor soil quality (e.g., high conductibility, low organic matter, etc.), and industrial pollution. On sandy loam soil (77% sand) with low organic matter content (1.2% organic matter) and low nutrients availability, the *A. donax* dry biomass yield was reported to be about 20 tha^−1^ [47]. This result was obtained with no irrigation, weeding or pest control. Taken together, these characteristics take this energy crop out of competition with food/feed cultures. Furthermore, *A. donax* has been classified as moderately salt tolerant with a 50% yield reduction at 11 dS m^−1^ salinity concentration [48]. Also, in this case, there is no competition with food-feed crops because, for these crops, the salinity concentrations determining 50% yield reduction are significantly lower, as in the case of corn (5.9 dS m^−1^) and rice (3.6 dS m^−1^) or similar as in the case of sugar cane (8–12 dS m^−1^). For sugar beets, it is reported a higher salt tolerance (15 dS m^−1^) but this culture requires a strong weeding control, irrigation and fertilization procedures. 

Marginal lands are growing worldwide due to anthropic activities, in fact secondary salinization affects 20% of irrigated land worldwide [49], and in Europe, the soil contaminated with heavy metals represent 6.24% (137,000 km^2^) of the total agricultural land [50]. This could mean an opportunity to cultivate energy crops environmental friendly such as *A. donax* in the near future.

The optimization of new strategy for the sustainable use of marginal lands in Mediterranean areas was the aim of the European project, OPTIMA (Optimization of Perennial Grasses for Biomass Production in the Mediterranean Area) [51]. This project, analyzing the production potential, in biomass terms, of four perennial species (miscanthus, giant reed, switchgrass and cardoon), together with other less known endemic species, highlighted the high adaptability and the high biomass production of *A. donax* when compared to the other energy crops [52,53,54].

## 2. Genetic Variability

Many studies in the last 20 years have investigated the genetic variability among *A. donax* populations, not only for taxonomic purposes, but also to evaluate the possibility of genetic improvement programs. Fingerprinting *A. donax* by molecular markers often showed low genetic differences even in populations with large areas of diffusion. The first studies run under “The European Giant Reed (*Arundo donax* L.) Network” consisted of RAPD (Random Amplification of Polymorphic DN) analysis, and found a low percentage of polymorphism among populations from Greece, Italy and southern France, clustered by their respective origins [23]. A subsequent paper reported a G/N (G = number of distinct genotypes, N = number of samples) diversity index of 0.460 by the analysis of 97 accessions in eight populations along the Santa Ana River (United States) by RAPD markers. These authors suggested that multiple introductions of different clones had been made into the area of that survey. Isozyme analysis also detected variations, even if they were slightly lower than the RAPD markers analysis [47].

Further analysis by 10 SRAP (Sequence Related Amplified Polymorphism) and 12 TE-based (Transposable element) primer combinations, on 185 accessions from a wider area in the United States, suggested genetic identity, and the authors hypothesized that even if multiple introductions are documented, the same clone could be growing in different parts of the world. The G/N diversity index reported was 0.011 in the United States and 0.050 in France [24]. A further investigation resulted in a Nei’s index of 0.0566 for Italian, 0.099 for Asian and Middle-Eastern, and 0.0744 for Mediterranean *A. donax* accessions [15].

Two years later, AFLP (Amplified fragment length polymorphism) fingerprinting scored the lowest genetic diversity for the species among 16 accessions in the Mediterranean area, with a Nei’s diversity index of 0.008 [28]. Higher diversity has been reported in an Australian study, where the investigation on three river systems led to 31 unique genotypes among 58 plant samples, with a G/N ratio of 0.815 [13]. Variability assessed with the biggest sample size of 362 accessions, mostly concentrated in the Rio Grande basin, Texas, Mexico and Spain, retrieved relatively high variability, with a higher Nei’s diversity index of 0.929 in Spain and 0.243 in north-central America. The same similar genotypic clusters can be found in both continents, and the conclusions of the authors are, as already pointed out by previous studies, that *A. donax* has been introduced several times in the areas of the survey [55]. The same set of primers, used for the molecular fingerprints of 15 Italian clones, revealed genetic clusters, with no exclusive correlation to geographical data, but shared among different sampling sites and a G/N of 0.933 [56]. Seven SSR primers randomly selected from Maize GDB, used to fingerprint a collection of 86 Italian *A. donax* accessions, revealed a low genetic diversity, with a Nei’s diversity index of 0.093 [57]. The genetic diversity among 31 accessions sampled in the United States, India and Nepal resulted in a G/N ratio of 0.81 and 21 distinguishable genotypes [58]. The next year, 218 accessions from Australia were fingerprinted by AFLP, identifying two groups through Nei’s identity test, one dominant as in previous studies, and genetic diversity of 1.5% within the groups. Somatic variation could be the source of this variability, even if the frequency of mutations for *A. donax* has not yet been reported [59]. As a synthesis of all the studies reviewed, the efforts to describe *A. donax* variability are summarized in Table 1.

The different values for genetic variability among *A. donax* clones reported in the other studies could be explained by considering the use of different molecular markers, population sizes and sampling techniques [60].

For example, higher scores resulted from studies on river systems or single countries and are often explained by multiple introductions. A lower variability is possible when a wider area but a limited number of accessions has been sampled [28]. Moreover, sampling from hydrogeological basins can maximize the probability of collecting *A. donax* genotypes that have adapted differently to environmental conditions [61].

In addition, the limited genetic variability is unlikely to explain the phenotypical variations among accessions, and epigenetic variations need to be considered. Flowering traits, for example, are positively correlated to the latitude of sampling sites, even years after transplanting, and a fast, epigenetic adaptation to climatic factors could be responsible for the high variability of these traits [56]. Considering the phenotypical variations reported, like variegation, no evidence of correlation with genetic differences has been detected by molecular markers’ analysis [24]. Epigenetic variability can compensate for the low genetic diversity and explain the great adaptability of this plant species. Phenotypical differences among ecotypes due to genetic and epigenetic differences were studied analyzing 96 accessions of *A. donax* collected from 14 different populations in Italy by a combined approach with AFLP and MSAP markers [62]. The genetic diversity highlighted was explained by subdivision of populations into two clusters, one including most of the mainland Italy accessions, and the other, samples collected in the Sardinia and Campania regions, suggesting two different introductions of *A. donax* L. and confirming a low genetic diversity. Pedo-climatic conditions may generate variations in DNA methylation status that drive the mechanisms of convergence and/or divergence of populations experiencing similar or dissimilar habitats [62]. Methylation patterns in response to stress may vary among ecotypes with tissue specificity. Considering the salt-tolerant ecotypes, “Canneto” and “Domitiana” exhibited a different methylation profile in roots and leaves in comparison with the more susceptible “Cercola” ecotype [63].

## 3. Clonal Selection

The clonal selection of *A. donax* L. was a strategy already used at the end of 1970s to improve the trait’s “cane number” [64]. A clone comparison was carried out in Italy in 1997 and 1998, utilizing springtime-transplanted rhizomes of 39 clones collected in the Sicily and Calabria regions. Different biometrical parameters were measured, in particular, stem density, stem weight and plant height, showing for each clone a positive correlation with biomass yield. These traits had a significant variance among clones, maintained during the two years of this study, with heritability (h^2^) of 0.23 for yield and 0.48 and 0.46 for, respectively, stem density and stem height [65]. Positive results in term of heritability in the first year also resulted from an investigation on eight clones, part of a 100 Italian *A. donax* collection, with a heritability index (h^2^) of 0.21 and 0.34 for stem height and stem diameter, respectively. As pointed out by the authors, differences may disappear over the years, considering that the heritability of certain traits could be related to epigenetic regulation [57,66].

Clonal selection recently gave results in terms of selection for propagation efficiency during field establishment. From the field-grown collection of the University of Milan of 100 Italian *A. donax* L. clones, nine clones were tested for hydroponic propagation, and thirty-two clones for in vitro shoot multiplication. Significant differences were reported among clones, considering the percentage of node buds sprouting in hydroponic propagation, which ranged from 19.6% to 71.7%, and the number of shoots produced in a 45-day cycle of tissue culture, which ranged from less than 10 to over 40. The clone Ad20 gave the best performance during hydroponic cultivation and had a high predisposition towards micropropagation [67].

Clonal selection has also been performed to select clones with relevant anthocyanin production (unpublished results). In fact, anthocyanin pigments are an easy, visible marker easy to obtain in a large clone collection. The germplasm collection managed by the University of Milan was evaluated during the regeneration phase, looking for the presence of red pigments. The clone Ad47 exhibited red tissues in 30% of the explants, while the other clones showed this trait with an average frequency of 15%. The Ad47 clone showed red pigmentation in different tissues, including callus, embryo, root, shoot and leaf (Figure 2). This result suggested that clonal selection could be used to improve/also modify secondary metabolites, or in general useful molecules, in *A. donax*.

The low tolerance to freezing of *A. donax* is one of the factors limiting its diffusion. Clonal selection through screening in controlled environment chambers is a viable method to select cold-resistant *A. donax* clones, to enlarge the possible cultivation areas. Low temperature exposure has been reported to increase the freezing tolerance of a Honduras *A. donax* ecotype, also resulting in increased soluble sugars and proline production [68].

Other environmentally limiting factors, such as drought and soil salinity, are met with a significant tolerance by *A. donax*, making this species suitable for producing energy on marginal lands, in a strategy of competition avoidance with food crops [69]. Calculation of the stress susceptibility index (SSI) based on physiological values, such as SLA, total dry weight, surface of green leaves and CO_2_ assimilation rate, has been used in a comparative study including five clones from the collection of the University of Catania, Italy, two clones from Spain, and one commercial clone from Germany. Water scarcity and excess of NaCl were combined or considered individually, for a two-month period, with measurements every 15 days of the physiological parameters in the different clones. Five salt-tolerant clones were selected, and a further experiment for the evaluation of their performance under mild and severe salt stress corroborated the results obtained. “Agrigento” was the most resistant clone to water stress and “Martinensis” to salt stress; “Martinensis” and “Piccoplant” were the most suitable for combined stress conditions, while “Fondachello”, “Cefalù” and “Licata” performed better under increasing salt levels [70]. In hydroponic conditions, severe treatment for 3 weeks under 150 mM NaCl can successfully discriminate salt-tolerant ecotypes, while a mild stress obtained with 50 mM did not succeed. Practical parameters that highlighted differences in stress tolerance include the fourth-leaf width and dry weight of shoots and roots. Among ten ecotypes tested with this method, “Sant’Angelo”, “Canneto” and “Nisida” had the highest leaf growth at day 7, while “Torre del Greco”, “Domitiana”, “Policastro” and “Cercola” had smaller leaves with significant reductions from control to salt treatment. “Torre Lama” had the highest shoot biomass production, followed by “Canneto”, “Domitiana and “Sant’Angelo”, and showed no influence of salt treatment on this parameter [54]. Comparing three Italian ecotypes from the region Campania, the most salinity tolerant ecotypes were “Domitiana” and “Canneto”, with low impact on growth performance in 3 weeks of cultivation, while “Cercola” exhibited a fast stress response with detrimental effects on growth [56]. 

A faster response of one clone sampled from dry environmental conditions, in terms of stomatal closure due to ABA diffusion, was highlighted in a comparison between Clone 6 and Clone 20 from southern Italy, even if other physiological measures were not significantly different in drought conditions [66]. Comparing two ecotypes, one Italian and one Bulgarian, the latter showed higher performance in drought stress conditions, due to metabolic adaption in terms of increased release of isoprene and a more consistent production of ROS-scavenging compounds such as flavonoids. On the other side, the Italian ecotype showed a lower ability to produce isoprene, resulting in higher oxidative pressure, with increased content of zeaxanthin and more severe effects on primary metabolism in conditions of mild drought stress [71]. In analogy with salt stress, mild water stress caused by rainfed conditions, and tested on three clones, sampled in Morocco and in the two Italian regions Sicily and Tuscany, did not succeed in highlighting qualitative traits for the improvement of *A. donax* tolerance to environmental stress. However, physiological and morphological variations among ecotypes can still be observed, in particular the Sicilian clone maintained its photosynthetic efficiency late into the growing season in comparison with the other clones [72]. The comparison of Moroccan with northern Italian ecotypes in drought conditions over 2 years, on physiological values, highlighted how adaption to drought could be flagged by a reduced stem density, considering that this trait of the Moroccan ecotype, was paired with a better root water uptake efficiency in 20–40 cm deep soil, in the second growing season, a higher leaf water potential, and increased xylem vessel area and water conductance in drought conditions. Moreover, this clone had reduced stem density and shallow roots density, when grown in these limiting conditions, while the Italian clone maintained its morphology unchanged [73]. Early stomatal closure in response to drought stress and recovery during subsequent watering had no discriminating value to evaluate *A. donax* ecotypes for water stress tolerance, but actual consideration of the Moroccan ecotype’s morphology, paired with its high ash content, could represent interesting selection measurements for improving productivity [74]. The comparison within a panel of 82 Euro-Mediterranean ecotypes, beside confirming phenotypical variations and heritability of 21 traits among clones, allowed the authors to associate ecotypes adapted to arid zones with early flowering and to cluster ecotypes based on this measure, providing additional tools for early evaluation of *A. donax* clone productivity [56,75].

Despite a vast literature on the phytoremediation potential of *A. donax* L., fewer publications have approached this topic in terms of clonal selection. Comparing two ecotypes, the American Blossom and the Hungarian 20 SZ, in in vitro embryogenic cultures with increasing amounts of sodium-selenate, differences among ecotypes for survival rate, selenium accumulation and growth parameters could be detected, especially at the concentration of 20 mg/L Na_2_SeO_4_. The Blossom ecotype exhibited higher selenium accumulation, with detrimental effects on growth performance [76]. On the base of these results, selection for target traits have to be carefully considered in the widest view possible, considering that a single positive parameter can negatively correlate to other key performance indices. In a similar setup, the copper tolerance, in the range of 0–26.8 mg L^−1^ CuSO_4_ pentahydrate, resulted in both ecotypes proving suitable for phytoremediation for this element [77]. Selection of clones suitable for lead and cadmium phytoremediation has been reached through sampling of ecotypes from ten populations of *A. donax* in non-ferrous and smelting areas of South China. Clones from Hunan and Yunan exhibited the highest accumulation factors for Pb and Cd, respectively, identifying these clones as promising for phytoremediation of these elements. This strategy suggested that a viable way for clonal selection with the purpose of phytoremediation is the in-situ evaluation of clones, considering the content of contaminants in the soil and some physiological parameters in *A. donax* plants, most notably the content of malondialdehyde [78].

## 4. Somaclonal Variation and Mutagenesis

To stably increment the genetic variability of *A. donax* besides observing the rise of epigenetic variants, somaclonal variation and mutagenesis methods must be considered. In a research project with our group, a large experimental field (5 ha) has been established in Uboldo, Italy, and four unique phenotypes have been identified among 15,000 individuals propagated via in vitro shoot culture. Among the four unique traits, three were variegated with a defective photosynthetic pathway (Figure 3), and one was a brachytic phenotype.

In a study on the results of embryogenic callus culture, variegation of *A. donax* plantlets was described, with 11 distinctive band pattern phenotypes reported out of 50,000 acclimated plantlets obtained from micropropagation, with white or clear green bands deficient in chlorophyll *a* and *b*, [79]. Based on these data, variegated phenotypes of *A. donax* obtained from regeneration through embryogenic callus or shoot culture, can occur with a frequency of approximately 1/5000. Variegated clones of *A. donax* L. were already reported 250 years ago [80] and were sold as ornamental plants in the United States [54]. Somaclonal variation can result in such cytological abnormalities but also in frequent qualitative and quantitative phenotypic mutations, sequence changes, and gene activation and silencing. Epigenetic mechanisms play a role in somaclonal variation, including activation of transposable elements, silencing of genes, and variating methylation pattern of single-copy sequences [81]. In *A. donax* L., transposable elements are estimated to represent 37.55% of the genome. The Ty3-*Gypsy* LTR-RT (Long Terminal Repeat-Reverse-Transcriptase) superfamily was the most represented, and estimated to represent 12.88% of the genome. Highly conserved copies of RIRE1-like Ty1-Copia elements are estimated at about 3%, while a majority remain unclassified and considered species-specific, not similar to any coding gene and not of plastidial origin [82]. During the in vitro development, the genome of the explants follows cycles of demethylation, while undifferentiating, and reaches different levels of methylation during regeneration. A consequence is that methylation at specific loci would decrease during dedifferentiation, and could not be re-established normally during regeneration [81]. In this perspective, important traits can be discovered or improved, and somaclonal variation represents a tool to increase and study *A. donax* variability. For example, in vitro somaclones tested for dehalogenation activity in roots extracts show higher variability than clones propagated vegetatively in nurseries. Application of near-lethal levels of trichlorophenol to in vitro cultures further increase the variability. As a measure of the effect of somaclonal variation on dehalogenation activity, the highest 95th and 99th percentile of somaclones increased, respectively, by 14% and 31% with no selective pressure, and by 64% and 93% with near-lethal levels of trichlorophenol [83].

Different methods for *A. donax* tissue culture, essential in the frame of its genetic improvement, have been investigated by different authors. Large scale shoot culture micropropagation of *A. donax* is rapidly achievable starting from stem nodes, preferably sampled in autumn, followed by subcultures on a modified Murashige and Skoog (MS) medium supplemented with BAP (benzylaminopurine) concentrations of 2–3 mg/L [84], 0.3 mg/L [85], 0.5–5 mg/L [86], 2.5–5 mg/L [87] or 1.25–5mg/L [67], with roots developing on the same shoot propagation medium. Successful acclimatization (>95% survival) of plantlets can be carried out, even in late winter, in a cold greenhouse or under simpler facilities such as shade nets [84]. Starting from a single explant, shoot organogenesis can produce up to 700 plantlets/year [86]. Embryogenic cultures can be obtained from immature inflorescences on a refined MS medium, that includes 2,4-D (2,4-dichlorophenoxyacetic acid), picloram, BAP, ZEA (zeatin) and TDZ (thidiazuron) growth regulators [87].

Young inflorescence segments can also be explanted and grown as callus cultures on medium supplemented with 05–5 mg/L of 2,4-D [88,89]. Another possibility is to utilize in vitro auxiliary buds, excised and cultured in dark conditions, on media supplemented with BAP 0–0.25 mg/L and 2,4-D 0–10 mg/L, with 3 mg/L as the optimal concentration that harmonizes organogenesis and callus formation, while BAP 0.25 mg/L alone is the best supplement to generate shoots [90]. The best material to initiate a callus culture, to our experience, is the basal node of in vitro micropropagated sprouts, followed by internode shoot sections. After initiation of the subculture, *A. donax* callus cultures can be maintained as undifferentiated tissue with up to 10 mg/L 2,4-D and 0.25–0.5 mg/L of BAP for 4–12 weeks long subcultures, also in light conditions. A method for callus preservation is cryo-conservation by one-step vitrification in liquid nitrogen, in vials containing a protective solution with MS salts, sugar, DMSO, glycerol and ethylene glycol. The loss of growth caused by the treatment is below 10% [91]. In most papers, totipotent calli of *A. donax* are often described as embryogenic calli, corresponding to light yellow-white calli, with granular friable consistency. As described in the more detailed papers, this totipotent tissue must be transferred to a secondary medium to effectively produce unipolar embryos [83]. Most papers do not report a clear distinction between organogenetic and embryogenetic regeneration, and this is notable, considering that the second is preferable in biotechnological improvement processes. Different media supplemented with different hormone combinations can be used to regenerate shoots and roots through the callus phase, using *A. donax* composite meristems and leaves. In our laboratory, undifferentiated callus cultures sub-cultured for 2 years on medium supplemented with 2,4-D 10mg/L and BAP 0.5 mg/L have been stimulated to regenerate on media supplemented with no growth regulators, 0.25 mg/L or 0.75 mg/L BAP. Measurement of explant-specific heterogeneity and semi-quantitative selection of regenerant callus lines may solve the problems limiting pro-embryogenic and organogenic regeneration from the callus phase (Figure 4). Only 7 out of 201 explants exhibited shoot regeneration, while roots were developed in the vast majority of explants. Well-developed embryo-like structures were obtainable by transferring pro-embryogenic callus lines onto a primary medium for embryogenesis [88].

Suspension cultures of *A. donax* could represent a powerful technique to exploit somaclonal variation, due to fast subcultures cycles and a higher increase in weight compared to solid medium, with considerable growth over 10–14 days. Liquid culture medium can contain a combination of 2–4,D and BAP or just glutamine as growth regulators [92]. Considering somatic embryogenesis to produce variegated variants for commercial use [79], embryogenetic culture, or a suspension culture before regeneration, should be preferred to reduce chimerism while increasing *A. donax* variability.

Besides this method, a more direct technique to increase variability in *A. donax* is mutagenesis by physical means. A ionizing radiation of 40–60 Gy is a suitable dose of gamma radiation to regenerate mutant clones from *A. donax* undifferentiated calli via organogenesis, considering a RD_50_ in the range of the dosage used to mutagenize other polyploid species. The frequency reported of almost 10% of aberrant phenotypes is quite high, suggesting that this technique is very promising to increase genetic variability in *A. donax.* Brachytic and dwarf phenotypes exhibited high heritability of these traits while variegated mutants showed different variegation levels among different shoots in the plants [93]. Physical mutagenesis by gamma radiation can induce modifications in biomass composition of *A. donax*, like increasing cellulose content to the levels of hardwood (41%), thus generating clones promising for second generation bio-ethanol production, but also reducing the Si/K ratio of biomass and so the ash melting point, which represents a detrimental trait for thermochemical conversion [94].

## 5. Genetic Engineering

So far, to our knowledge, no transgenic *A. donax* plants with improved characteristics have been developed, possibly due to the limited regeneration of tissues and the absence of traits of interest well characterized at the molecular level. In any case, transient expression of GUS and GFP reporter genes is obtainable by an optimized particle bombardment protocol on *Arundo donax* callus cells. Important parameters to be taken into account include helium pressure, distance from stopping screen to target tissue, value of vacuum pressure, material and size of the microparticles, DNA concentration and number of bombardments. Higher efficiency in DNA transfer, resulting in 100–150 modification positive spots for explant, is achievable with cells bombarded twice at 1100 psi, with 9 cm target distance, 24 mm Hg vacuum pressure, 1 mm gold particle size, 1.5 µg DNA per bombardment, three days of pre-culture before the bombardment and six days of culture after bombardment. Bombardment with a GFP reporter gene resulted in higher expression than using GUS gene. The 35s promoter of CaMV can be used for the constructs, with hygromycin resistance to select modified cells [89]. Transformation of *A. donax* is also possible through protoplast manipulation. Starting from suspension culture, cell walls can be digested with a solution containing food-grade enzymes and 2–3 h incubation at 37 °C. The comparison among promoters highlighted that Ubi2 promoter from *P. virgatum* L. is a stronger promoter than CaMV 35S, with the second inducing low expression in *A. donax*. Trials with a different set-up of PEG-mediated transformation failed, while electroporation carried out at 130 V and 1000 μF resulted in a transformation efficiency of 3.3% ± 1.5% [92].

The study of the *A. donax* genome is still a demanding research topic aimed at collecting knowledge needed for *A. donax* genetic improvement, since the lack of a high-quality reference genome sequence. A hybrid approach combining Illumina and long-read sequencing technologies, i.e., Pac Bio or Nanopore, could be used as previously reported for de novo sequencing in other crops [95,96]. In particular, the high-quality reference genome of *Oryza longistaminata* has been obtained incorporating Illumina and PacBio sequencing data [95], while the de novo genome sequence assembly of trifoliate yam (*Dioscorea dumetorum*) was the result of Illumina and Oxford Nanopore technologies [96].

A parallel approach to identify putative target genes for A. donax genetic improvemt is the utilization of *S. italica* genome, the more related species that is actually sequenced. In fact, considering the lignin biosynthetic pathway genes, in particular PAL-like and *CCoAMT*-like genes, the high homology of four Mediterrean ecotypes of *A. donax* transcripts with *S. italica* L. [90] represent an important perspective for mining possible target sequences about this trait improvement by bioinformatics. A similar approach has been recently applied to isolate potential gene target to be used for genetic improvement of Miscantus × giganteus, a promising lignocellulosic biomass crop for biofuel production. Transcriptional analyses and phylogenetic and genome synteny analyses have allowed the identification of the major monolignol biosynthetic genes and the putative transcription factors regulating their expression [97] (Zeng et al. 2020).

Different reviews have been published on genetic improvement of energy crops, but for the energy crops suitable for genetic transformation, such as swichgrass, important results in the improvement of energy production have been obtained by gene silencing [98,99] and CRISPR/CAS 9 (Clustered Regularly Interspaced Short Palindromic Repeats/CRISPR associated protein 9) [100] techniques. However, the data reported for *A. donax* are based on the results of agronomic studies with the aim of optimization of biomass quality.

Currently, sequence databases and information on organs diversity and possible targets for improvement can be found in RNA-seq Illumina transcriptomics studies. Gene Ontology Analysis of metabolic differences among bud, culm, leaf and root tissues highlighted that the most variety can be found in the leaves, most notably for light, osmotic, salt and metal stress response, and for primary and secondary metabolites production [101]. About 40–45% of transcripts showed homologies with known sequences and functional annotations of *Oryza sativa* L., *Triticum aestivum* L. and mostly with *S. bicolor* L. and *Z. mays* L., most importantly for gene categories related to flowering time, plant height and structure, carbohydrates composition and vernalization response. *CCoAMT*-like genes deserve particular attention for their possible role in obtaining mutants with decreased content of lignin in culms [101]. The response of *A. donax* to low oxygen stress analyzed by a metabolomic approach [102] provided numerous insights required to target functional genes by transcriptomics.

About the well-known *A. donax* tolerance to low soil quality, RNA-seq provided insights of the available defenses from adverse soil conditions. Considering the excess of Ni and Cu, doses of 25–100 mg/L activated the expression of a metal-uptake *YSL*-like gene and a macrophage protein which was *NRAMP*-like [103]. Adjustments in phytochelatin synthases expression could represent a reliable strategy to increase *A. donax* uptake of metalloid contaminants with the purpose of phytoremediation. With an RNA-seq methodology, three putative genes, *AdPCS1-3* have been identified in *A. donax*. The expression of these three genes in response to CdSO_4_ stress was tissue specific, with *AdPCS1* the most up-regulated compared with control. However, the production of *Arabidopsis thaliana* L. transgenic lines overexpressing these genes resulted in deleterious effects on growth, with necrotic effect, while the same strategy applied to yeast resulted in Cd-tolerant lines [104]. The responses to salt stress and salt tolerance are other important traits investigated by RNA-seq with improvement purposes. Different ecotypes exhibited a possible positive correlation of salt exposure with the expression of stress-induced transcription factors *DREB2A*-like and *WRKY53*-like, activation of detoxification processes and abscisic acid increase. Moreover, a fast response to salt stress, with overexpression of ion transporters and K+/Na+ homeostasis-related genes, such as *SOS1*-like, *NHX1*-like or *KHT1*-like, represents an effort to reduce the ionic stress, but was detrimental to the growth performance [63,105]. A RNASeq analysis [106] conducted under long-term salt stress allowed the identification of differentially expressed genes with a dose-dependent response. The analysis was performed on a total 38,559 DEGs (differentially expressed genes) and among them, 2086 were up-regulated and 1766 were down-regulated.

In this paper, in particular, it is reported the analysis of clusters related to salt sensory and signaling, hormone regulation, transcription factors, Reactive Oxygen Species (ROS) scavenging, osmolyte biosynthesis and biomass production. Several unigenes identified have the potential to be used to improve productivity and stress tolerance in *A. donax*. In particular, the silencing of the GTL1 gene (a homolog of Setaria italica trihelix transcription factor) acting as a negative regulator of water use efficiency could be a good target for NBT (new breeding techniques).

## 6. Conclusions

The aim of this review was to emphasize the potential genetic methodologies to improve *Arundo donax*, an emerging perennial crop plant for biomass production and industrial applications and to discuss the main challenges and putative solutions.

In Figure 5, we have summarized the main targets of short-to-medium-term *A donax* genetic improvement. In particular, we considered four steps regarding the complete *A. donax* pipeline: plant production, stand establishment, crop growth and biomass processing.

Literature data suggest that clonal selection is the best effective method up to now for selection towards yield [57,65], in vitro propagation efficiency [67], salt tolerance [61,63,70] and phytoremediation [76,77,78]. Physical/chemical mutagenesis [93] and somaclonal variation [79,83] could represent opportunities to generate new genetic variability, although, since *A. donax* L. is a polyploid species, it may be very difficult to genetically modify it and overcome the redundancy of genetic information.

Due to the low genetic variability found in *A. donax* L. wild populations, genetic engineering could represent an alternative solution to directly modify the expression of specific genes, making *A. donax* even more competitive than other energy crops. In fact, *A. donax* L. transformation [89,92] is possible, although so far, it is not a consolidated method due to the low regeneration efficiency from callus cultures.

Once efficient protocols to overcome these technical limits become available, the recent advances in *A. donax* L. molecular markers [13,15,55,57,58] and transcriptomics [63,101,103,105,107] could be used to improve several important traits of this energy crop, for example herbicide resistance [108] and lignin content [109]. New genome editing techniques like CRISPR/cas9, are waiting to be applied for straightforward improvement of a wide range of selected traits, overcoming possible genetic redundancy caused by polyploidy. However, the potential of *A. donax* L. for energy production and industrial applications is becoming established, and genetic improvement of this species will represent an important part in the new deal of the green economy.

## Figures and Tables

**Figure 1 plants-09-01584-f001:**
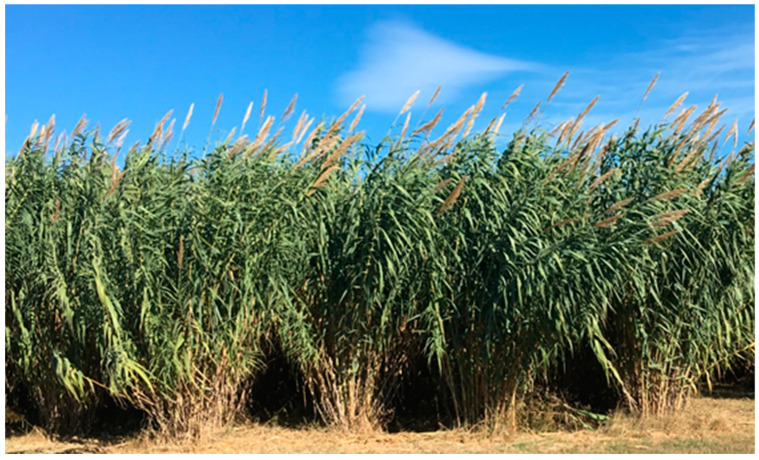
Crop field of *A. donax* L. for energy purposes in the third year of cultivation.

**Figure 2 plants-09-01584-f002:**
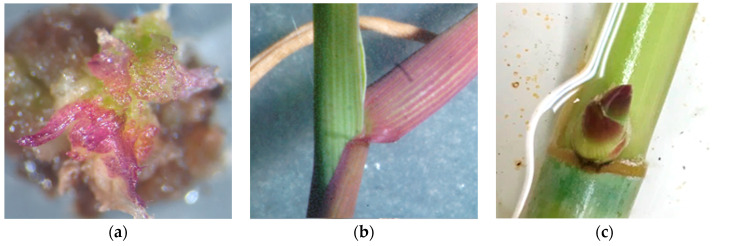
Accumulation of anthocyanins in different tissues of Ad47 clone. (**a**) Embryogenic callus grown under circadian light cycle. (**b**) Shoot regenerated from organogenic callus. (**c**) Swollen bud from hydroponic propagation of vegetatively active canes.

**Figure 3 plants-09-01584-f003:**
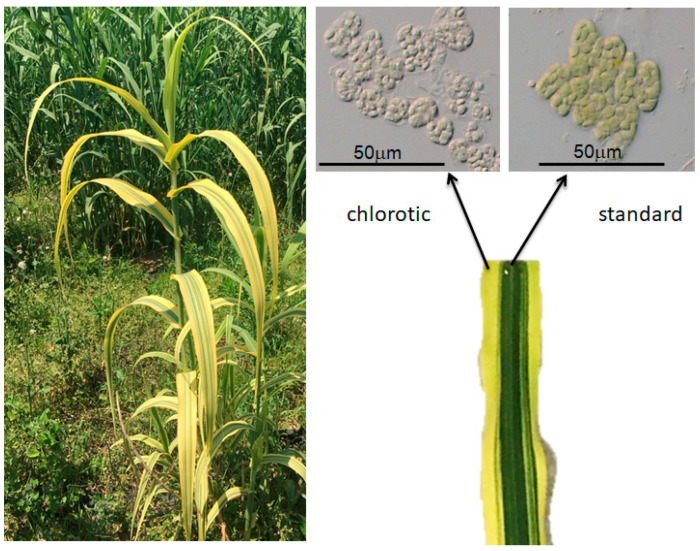
Phenotype on plant-scale and at histological level of variegated *A. donax* L. accessions. These individuals have reduced growth, and the chlorotic yellow to white striped sectors exhibit defective chloroplasts, with no accumulation of chlorophylls and reduced membrane organization and dimensions.

**Figure 4 plants-09-01584-f004:**
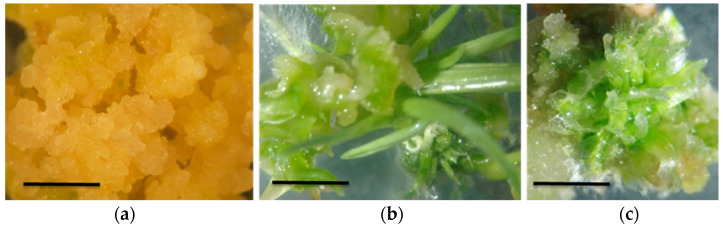
*A. donax* L. micropropagated calli with different regeneration aptitude: (**a**) pro-embryogenic undifferentiated callus, (**b**) organogenic callus, (**c**) embryogenic callus. Scale bars indicate 5 mm in (**a**), 10 mm in (**b**) and (**c**).

**Figure 5 plants-09-01584-f005:**
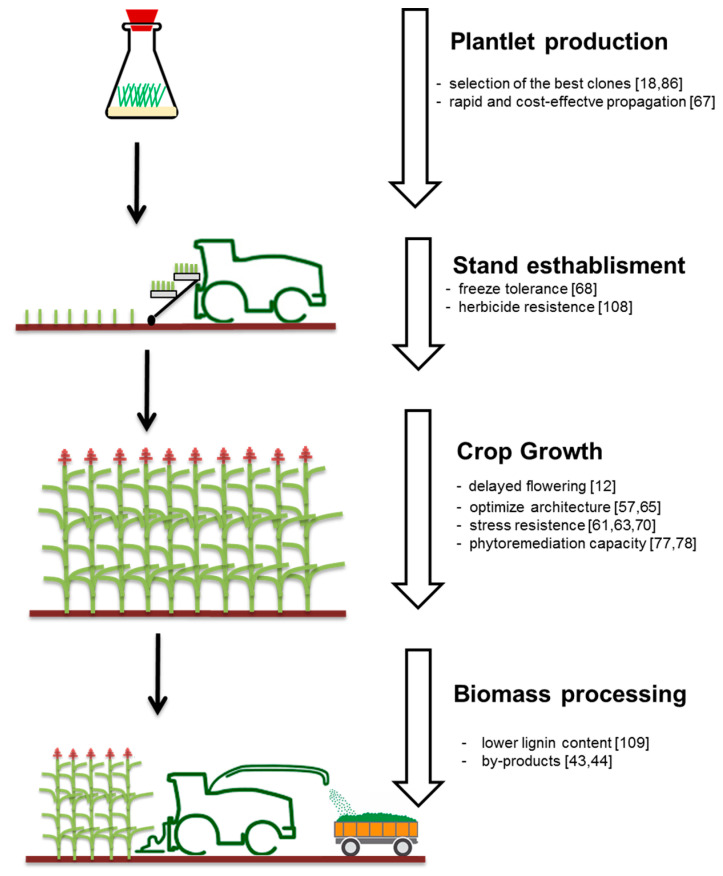
Graphic representation of feasible genetic improvements for *A. donax* L. Targets are categorized by crop phases and coupled with the significant references.

**Table 1 plants-09-01584-t001:** Genetic difference among *A. donax* L. accessions using different molecular approaches expressed as G/N ratio (G = number of distinct genotypes, N = number of samples) or Nei’s index.

Sample Size	Markers (n)	Location	G/N	Nei’s Index	Reference
97	RAPD (14)	California	0.460	-	[54]
	Isozyme (2)	USA	0.092	-	[54]
185	SRAP (10)	USA	0.011	-	[24]
	SRAP (10)	France	0.050	-	[24]
12	ISSR (10)	Italy	0.083	0.0566	[15]
122	AFLP (6)	Asia-Middle East	-	0.099	[15]
	AFLP (6)	Mediterranean	-	0.0744	[15]
16	AFLP (6)	Mediterranean	-	0.008	[28]
58	ISSR (10)	Australia	-	0.815	[13]
159	ISSR (10)	Mexico-USA	-	0.243	[55]
203	ISSR (10)	Spain	-	0.929	[55]
15	ISSR (10	Italy	0.933	-	[56]
86	SSR/STS (7)	Italy	0.093	-	[57]
31	ISSR (20)	USA-India-Nepal	0.81	-	[58]
218	AFLP	Australia	0.94	0–0.192	[59]

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
