# Peer review of "Genetic Improvement of *Arundo donax* L.: Opportunities and Challenges"

_plants, 2020, doi:10.3390/plants9111584_

Round 1
Reviewer 1 Report
This review by Danelli et al., addresses important topics, has a good structure and is well written. Please see my comments below for potential aspects to further improve the strengths of this manuscript.
Major comments:
1) Genetic improvements should harness the power of cutting-edge genomics to boost SMART breeding. Additionally, the genetic engineering of this crop would require knowledge about the genome. For example, tissue specific promoters might be needed for most applications and CRISPR/Cas-based approaches would require confirmation of desired mutations. It might be interesting for future research to discus different options or a strategy for the generation of a high quality reference genome sequences. Would it be possible to use cost-effective ONT sequencing (like doi:10.3390/genes11030274) or would PacBio (like doi:10.1002/tpg2.20001) be a better approach? Are populations available to produce genetic maps for scaffolding of contigs? Which genotype/line should be selected for the generation of such a reference genome sequence?
2) How does A. donax compare to other crop species like Beta vulgaris with respect to salt tolerance? I think this would be interesting for readers in the context of "competition avoidance".
3) Since the authors mention transcriptomic data sets, it could be possible to identify candidates in the anthocyanin biosythesis. An automatic identification is possible which would make this relatively easy. Over 50 publicly available RNA-Seq datasets would even allow quantification in different tissues/ under different conditions.
4) The authors might want to provide some details about regions where A. donax farming would be feasible: 1) with respect to food competition avoidance, 2) for phytoremediation, and 3) genetic engineering applications. I think that the last one would at least exclude Europe.
Minor comments:
line 102: TE-based primers combinations >TE-based primer combinations
The paragraph about genetic diversity could be written more concisely. There are some details about the cited studies which could be ommitted here.
Table 1: why is "SSR/STS(7)" underlined? The caption of this table should explain why there is no Nei's index in the last row.
line 186: anthocyanins production > anthocyanin production
Fig. 2: Is anthocyanin production a response to cold stress? Would it be possible to use anthocyanins as a screening marker for genetic engineering applications of A. donax?
line 262: 0-26.8mg L -1 CuSO 4 > 0-26.8 mg L-1 CuSO4 (space missing)
line 266: for, respectively, Pb and Cd, > for Pb and Cd, respectively,
line 286: frequency of 1/5000 > frequency of approximately 1/5000
The TE proportion estimates are surprisingly precise given the little knowledge about the genome sequence.
line 310/line 316: BAP, ZEA ...? Please explain the hormone abbreviations.
line 314: There seems to be something wrong with reference 79.
Fig. 4: Please add scale bars.
line 423: Is there a difference between selenium accumulation and phytoremediation?
Fig. 5: erbicide resistance > herbicide resistance
line 505: Please check the formatting of reference 19.
It looks like there are double spaces in some places. Please search and replace these.
Author Response
Reviewer 1
Major comments:
- Genetic improvements should harness the power of cutting-edge genomics to boost SMART breeding. Additionally, the genetic engineering of this crop would require knowledge about the genome. For example, tissue specific promoters might be needed for most applications and CRISPR/Cas-based approaches would require confirmation of desired mutations. It might be interesting for future research to discus different options or a strategy for the generation of a high quality reference genome sequences. Would it be possible to use cost-effective ONT sequencing (like doi:10.3390/genes11030274) or would PacBio (like doi:10.1002/tpg2.20001) be a better approach?
The aim of this review was to present the efforts regarding the genetic improvement of this promising energy crop that could be applied in short -middle term. Due to the fact to our knowledge no A donax GMO was created because to the difficulties in regenerating transformant cells to plantlets we decided to concentrate our efforts on techniques that can be used in middle term. For this motif we have not “stressed” the transcriptomic/genomic data. However, to address your issue we added and discussed other papers regarding genomic and transcriptomic data.
Are populations available to produce genetic maps for scaffolding of contigs? Which genotype/line should be selected for the generation of such a reference genome sequence?
Arundo donax is a sterile plant and of course it is impossible to perform any type of crosses among different clones, hence we cannot produce any segregating population or other genetic material useful for mapping or positional cloning approaches. To our knowledge, as reported by Ardion’s papers, most probably A. donax was selected by human in Indu valley, reached Mediterranean basin and then spread around the world. Due this monophyletic origin the molecular differences among different clones are low, then, of course, each clone would be used as source for genomic sequencing.
- How does A. donax compare to other crop species like Beta vulgaris with respect to salt tolerance? I think this would be interesting for readers in the context of "competition avoidance".
We added some considerations regarding the competition between energy and food-feed crops, in particular regarding the utilization of marginal lands because of the low fertility and high conductibility.
- Since the authors mention transcriptomic data sets, it could be possible to identify candidates in the anthocyanin biosynthesis. An automatic identification is possible which would make this relatively easy. Over 50 publicly available RNA-Seq datasets would even allow quantification in different tissues/ under different conditions.
We just used anthocyanin pigments as visible markers easy to score in a large clones collection to prove that clonal selection could be used to improve/modify also secondary metabolites or in general useful molecules, in A donax. Anthocyanins are not a target of the A. donax genetic improvement.
- The authors might want to provide some details about regions where A. donax farming would be feasible: 1) with respect to food competition avoidance, 2) for phytoremediation, and 3) genetic engineering applications. I think that the last one would at least exclude Europe.
We add this consideration in Introduction:
“The capacity of A. donax to grow on marginal and abandoned lands makes this plant able to be cultivated on soil not suitable for traditional agriculture. Lands can be defined as marginal for different reasons, as water scarcity, poor soil quality (e.g. high conductibility, low organic matter, etc…), industrial pollution. On a sandy loan soil (77% sand) with low organic matter content (1.2% organic matter) and low nutrients availability, the A. donax dry biomass yield was reported to be about 20tha-1 [47]. This result was obtained with no irrigation, weeding and pest control. Taken together these characteristics make this energy crop not competing with food/feed cultures. Furthermore, A donax has been classified as moderately salt tolerant with a 50% yield reduction at 11,4 dS m-1 salinity concentration [48]. Also in this case there is no competition with food-feed crops due to the fact that for these crops the salinity concentrations determining 50% yield reduction are significantly lower, as in the case of corn ( 5.9 dS m-1 ) and rice (3,6 dS m-1) or similar as in the case of sugar cane (8-12 dS m-1). For sugar beet it is reported a higher salt tolerance (15 dS m-1) but this culture requires a strong weeding control, irrigation and fertilization procedures.
Marginal lands are growing worldwide due to anthropic activities, in fact secondary salinization affects 20% of irrigated land worldwide [49], and in Europe the soil contaminated with heavy metals represent 6,24% (137000 km2) of the total agricultural land [50]. This could represent an opportunity for the cultivation of energy crops environmental friendly such as A. donax in the next future.
The optimization of new strategy for the sustainable use of marginal lands in Mediterranean areas was the aim of the European project OPTIMA (Optimization of Perennial Grasses for Biomass Production in the Mediterranean Area) [51]. This project, analysing the production potential, in biomass terms, of four perennial species (miscanthus, giant reed, switchgrass and cardoon) together with other less known endemic species, highlighted the high adaptability and the high biomass production of A. donax when compared to the other energy crops [52,53].”
Minor comments:
line 102: TE-based primers combinations >TE-based primer combinations
modified as suggested
The paragraph about genetic diversity could be written more concisely. There are some details about the cited studies which could be ommitted here.
As suggested, we deleted some details and we shortened the paragraph
Table 1: why is "SSR/STS(7)" underlined? The caption of this table should explain why there is no Nei's index in the last row.
We corrected table 1: we eliminated the underlining, we added the Nei’s index in the last row and the requested information in caption.
line 186: anthocyanins production > anthocyanin production
modified as suggested
Fig. 2: Is anthocyanin production a response to cold stress? Would it be possible to use anthocyanins as a screening marker for genetic engineering applications of A. donax?
It is well known that cold affects anthocyanin accumulation in different tissues, and also in Arundo we can push anthocyanin accumulation by cold treatment. Starting from our clone collection we selected a clone (Ad47) able to accumulate more pigments with respect to the other clones, without any treatment. Of course by cold treatment anthocyanin accumulation can be improved. For what concern the possible application of anthocyanin as screening marker for genetic engineering applications we think that it could be a useful tool to select transformed cells avoiding the utilization of “most impacting” markers such as antibiotic resistance.
line 262: 0-26.8mg L -1 CuSO 4 > 0-26.8 mg L-1 CuSO4 (space missing)
Corrected as suggested
line 266: for, respectively, Pb and Cd, > for Pb and Cd, respectively,
Corrected as suggested
line 286: frequency of 1/5000 > frequency of approximately 1/5000
Modified as suggested
The TE proportion estimates are surprisingly precise given the little knowledge about the genome sequence.
We reported the data presented in :
Lwin, A.K.; Bertolini, E.; Pè, M.E.; Zuccolo, A. Genomic skimming for identification of medium/highly abundant transposable elements in Arundo donax and Arundo plinii. Molecular Genetics and Genomics 2016, 292(1), 157–171.
line 310/line 316: BAP, ZEA ...? Please explain the hormone abbreviations.
We added the full name after each abbreviation at first mention
line 314: There seems to be something wrong with reference 79.
We checked and corrected reference 79
Fig. 4: Please add scale bars.
We added the scale bars.
line 423: Is there a difference between selenium accumulation and phytoremediation?
Of course not, we modified the sentence:
Literature data suggest that clonal selection is the best effective method up to now for selection towards yield [50,58], in vitro propagation efficiency [60], salt tolerance [54,56,63] and phytoremediation [69,70,71].
Fig. 5: erbicide resistance > herbicide resistance
Modified as suggested
line 505: Please check the formatting of reference 19.
It looks like there are double spaces in some places. Please search and replace these.
We checked the spaces
Reviewer 2 Report
This review analysed the recent advances and challenges encountered in using clonal selection, mutagenesis/somaclonal variation and transgenesis/genome editing. Attempts to improve crop yield, in vitro propagation efficiency, salt and heavy metal tolerance by clonal selection were carried out, although limited by the species’ low genetic diversity and availability of mutants. Authors highlight the potential genetic methodologies to improve Arundo donax, an emerging perennial crop plant for biomass production and industrial applications and to discuss the main challenges and putative solutions. The topic is within the scope of this journal.
- Arundo donax should be italics in full MS.
- Abstract: Abstract is not clear. Please improve the abstract section.
- I feel lack of information of Genetic improvement of Arundo donax in the MS. Lot of works were published in Genetic improvement of Arundo donax. Please updates all genetic improvement.
- Overall, this MS need to be improved.
- Please update more genetic improvements in Arundo donax . Figure 5. Graphic representation of feasible genetic improvements for donax.
- Conclusion: Conclusion is not clear. Please improve the abstract section.
Author Response
Reviewer2
This review analysed the recent advances and challenges encountered in using clonal selection, mutagenesis/somaclonal variation and transgenesis/genome editing. Attempts to improve crop yield, in vitro propagation efficiency, salt and heavy metal tolerance by clonal selection were carried out, although limited by the species’ low genetic diversity and availability of mutants. Authors highlight the potential genetic methodologies to improve Arundo donax, an emerging perennial crop plant for biomass production and industrial applications and to discuss the main challenges and putative solutions. The topic is within the scope of this journal.
Arundo donax should be italics in full MS.
We checked all the text and modified as suggested
Abstract: Abstract is not clear. Please improve the abstract section.
The Abstract section has been improved as suggested.
I feel lack of information of Genetic improvement of Arundo donax in the MS. Lot of works were published in Genetic improvement of Arundo donax. Please updates all genetic improvement.
The aim of this review was to present the efforts regarding the genetic improvement of this promising energy crop that could be applied in short -middle term. Due to the fact to our knowledge no A donax GMO was created because to the difficulties in regenerating transformant cells to plantlets we decided to concentrate our efforts on techniques that can be used in middle term. For this motif we didn’t stressed the transcriptomic/genomic data. However, to address your issue we added and discussed other papers regarding genomic and transcriptomic data.
Overall, this MS need to be improved.
As suggested, we improved all the sections of this manuscript.
Please update more genetic improvements in Arundo donax . Figure 5. Graphic representation of feasible genetic improvements for donax.
Conclusion: Conclusion is not clear.
As suggested, we improved the section Conclusion.
Round 2
Reviewer 2 Report
Requested corrections were completed.